# High Dose Thoracic Re-Irradiation and Chemo-Immunotherapy for Centrally Recurrent NSCLC

**DOI:** 10.3390/cancers14030573

**Published:** 2022-01-23

**Authors:** Brane Grambozov, Markus Stana, Bernhard Kaiser, Josef Karner, Sabine Gerum, Elvis Ruznic, Barbara Zellinger, Raphaela Moosbrugger, Michael Studnicka, Gerd Fastner, Felix Sedlmayer, Franz Zehentmayr

**Affiliations:** 1Department of Radiation Oncology, Paracelsus Medical University, SALK, A-5020 Salzburg, Austria; b.grambozov@salk.at (B.G.); m.stana@salk.at (M.S.); j.karner@salk.at (J.K.); s.gerum@salk.at (S.G.); e.ruznic@salk.at (E.R.); g.fastner@salk.at (G.F.); f.sedlmayer@salk.at (F.S.); 2Department of Pneumology, Paracelsus Medical University, SALK, A-5020 Salzburg, Austria; bernhardkaiser@gmx.at (B.K.); r.moosbrugger@salk.at (R.M.); m.studnicka@salk.at (M.S.); 3Institute of Pathology, Paracelsus Medical University, SALK, A-5020 Salzburg, Austria; b.zellinger@salk.at

**Keywords:** lung cancer, re-irradiation, immunotherapy, NSCLC, stage III

## Abstract

**Simple Summary:**

Since the early 1980s, there has been a trend towards escalating radiation doses in pulmonary tumor recurrences with the aim of improving survival. In this context, we performed a literature search in order to summarize the evidence of curative thoracic re-irradiation for centrally recurrent lung cancer. Tumor relapse in this specific situation poses a major problem because of the proximity to mediastinal organs. Of the initial 227 studies, 11 fulfilled the inclusion criteria for this analysis. The median overall survival (mOS) was 18.1 months (range 9.3–25.1), the median progression-free survival (mPFS) was nine months (range 4.5–16), and the median locoregional control (mLRC) was 12.1 months (range 6.5–20). The total re-irradiation dose correlated with both mLRC (*p*-value = 0.012) and mOS (*p*-value = 0.007). As large-scale prospective trials in the field are missing, this literature review is primarily based on retrospective data. In today’s age of enhanced long-term survival rates after chemoradiotherapy followed by immune checkpoint inhibition, the current analysis provides valuable insights into radiation treatment options for patients with loco-regional lung cancer recurrence.

**Abstract:**

Introduction: Thoracic re-irradiation for recurrent lung cancer dates back four decades, when the first small series on 29 patients receiving palliative doses was published. With 5-year overall survival rates of 57% in PDL-1 positive patients after primary chemo-radio-immunotherapy, the number of patients who experience loco-regional relapse will increase in the near future. In this context, centrally recurring lung tumors pose a major treatment challenge. Hence, the aim of the current review is to compile the available evidence on curatively intended thoracic re-irradiation for this special clinical situation. Methods: A systematic literature search according to the PRISMA guidelines was performed. A study was included when the following criteria were met: (1) 66% of the patients had NSCLC, (2) a total dose of 50 Gy in the second course and/or a biologically effective dose of at least 100 Gy in both treatment courses was administered, (3) re-irradiation was administered with modern radiation techniques, (4) 50% or more of the patients had a centrally located relapse, (5) the minimum cohort size was 30 patients. Results: Of the initial 227 studies, 11 were analyzed, 1 of which was prospective. Median overall survival (OS) was 18.1 months (range 9.3–25.1), median progression free survival (PFS) was nine months (range 4.5–16), and median loco-regional control (LRC) was 12.1 months (range 6.5–20). Treatment-related mortality rates ranged from 2% to 14%. The total dose at re-irradiation correlated with both LRC (*p*-value = 0.012) and OS (*p*-value = 0.007) with a close relation between these two clinical endpoints (*p*-value = 0.006). The occurrence of acute toxicity grade 1 to 4 depended on the PTV size at re-irradiation (*p*-value = 0.033). Conclusion: The evidence regarding curative re-irradiation for centrally recurrent NSCLC is primarily based on scarce retrospective data, which are characterized by a high degree of heterogeneity. The OS in this clinically challenging situation is expected to be around 1.5 years after re-treatment. Patients with a good performance score, younger age, small tumors, and a longer interval to recurrence potentially benefit most from re-irradiation. In this context, prospective trials are warranted to achieve substantial advances in the field.

## 1. Introduction

Thoracic re-irradiation dates back several decades, with reports mainly focusing on low-dose palliation. In 1981, Green published the first series of 29 patients, 20 of whom showed some type of response to thoracic re-irradiation with 20–40 Gy [1]. At present, the loco-regional relapse rates after radical treatment amount to 20–44% [2,3,4,5]. Since the successful introduction of immunotherapy in the primary setting, a median overall survival (mOS) of 57 months has been achieved in PDL-1-positive patients [6,7], which will lead to an increase in the number of patients with locoregional relapses.

In this regard, treatment options are still limited with a median overall survival (mOS) rate of 10–12 months after palliative chemotherapy (CT) in patients without mutations [8], which resembles initial stage IV disease [2,9]. The first choice for local therapy in these mostly inoperable patients is locoregional RT. Overall, palliative thoracic re-irradiation leads to reasonable symptom control in 70% of the patients [8,9]. A special clinical situation in this context is the occurrence of central tumors, which by definition are located 2 cm within mediastinal and hilar structures [10]. Even with technical advances in recent years, such as intensity-modulated radiotherapy (IMRT), stereotactic ablative body radiotherapy (SABR), protons, image-guided radiotherapy (IGRT), and Monte-Carlo based dose calculation algorithms, these centrally located tumor recurrences are regarded as more challenging to treat than recurrences elsewhere in the thorax due to their proximity to dose-limiting organs at risk (OAR). The variety of treatment schedules and techniques used in the first and second treatment course usually make a direct dosimetric comparison difficult. Hence, a more suitable measure for dose is the biologically equivalent dose in 2 Gy fractions (EQD_2_). In a recent review, curatively intended high-dose re-irradiation was defined as the application of a second radiotherapy course with total physical doses of 66 Gy [11], resulting in cumulative EQD2s that exceeded the generally accepted dose limits for the organ at risk [12]. At the same time, some authors consider a cumulative EQD_2_ > 100 Gy as too toxic for patients with centrally located tumors because of potential radiation-induced damage to the esophagus and blood vessels [8].

When discussing radical RT as a second course of treatment, the question of biological effectiveness arises. The concept of insufficient DNA damage repair may help to better understand the mechanism at work [13]. In theory, a first course of irradiation causes double strand breaks (DSBs) and clustered damages, which saturate both non-homologous endjoining (NHEJ) and homologous recombination (HR) pathways in a tumor cell. If in such a situation a second course of irradiation is applied, the additional dose may lead to further DNA damage, which overpowers the already strained repair potential and consequently leads to increased cell death. In this sense, re-irradiaton is a two-edged sword, since the effects on DNA repair in a tumor cell are reflected in normal tissue, which is of major concern since recovery between high dose treatments is not very well understood [14,15].

Therefore, if re-irradiation is intended to be curative, outcome, toxicity, and predictive/prognostic factors must be considered for adequate patient selection in order to balance benefit and risk of this approach. An overview of the relevant literature in the field is encumbered by the fact that most studies are small and retrospective in design. One of the very few exceptions is a prospective trial on re-irradiation with protons [3], which concluded that patients with good performance status and small tumor volume at recurrence benefit most from re-RT [3]. The impact of modern systemic treatment options, such as immunotherapy and tyrosine-kinase inhibitors (TKI), administered sequentially to re-irradiation is still being investigated. Immune checkpoint inhibitors (ICI) work via several membrane receptors resulting in the activation of downstream target proteins in the PI3K/AKT-, RAS/MEK/ERK-, and STAT-pathways. The clinically most relevant receptors are the cytotoxic T-lymphocyte-associated protein 4 (CTLA-4) and programmed cell death protein 1 (PD-1) as they can be blocked by targeted therapies which are currently used for various malignant tumor types, including lung cancer. PD-1, which has two ligands (PDL-1 and PDL-2) is expressed on T-, NK-, and B-cells [16]. Immune checkpoint inhibitors (ICI) such as Nivolumab, Atezolizumab, Pembrolizumab, and Durvalumab, which are now widely used for the treatment of lung cancer, block PD1/PDL-1 binding and thereby enable recognition of tumor cells by activated CD8+ T cells. Of note, although one meta-analysis showed that ICIs are equally effective in male and female patients [17], some clinical studies suggest a gender difference in response to ICI treatment for lung cancer (reviewed by Conforti [18]). The underlying mechanistic explanation could be an estrogen-induced upregulation of PD1-/PDL-1, as described in rodents [19]. Additionally, tumor cells with deficient repair enzymes show increased genomic instability (GI), which results in better response to ICI. In this context it should be emphasized that radiation also enhances GI paving the way for combined treatment with ICI in case of recurrent disease. The fact that chemotherapy alone in recurrent tumors without driver mutations, such as EGFR and KRAS, shows response rates of 20% to 35%, highlights the role of a potentially curative therapeutic option such as thoracic re-irradiation. On top of that, targeted drugs may either harbor the potential for concurrent treatment or enlarge the timespan between radiation treatment courses. Of note, it seems that patients with EGFR or KRAS mutations do not benefit from ICI treatment (reviewed by Xu Yangyang [17]).

The aim of this review is to summarize the evidence on clinical outcome and toxicity for patients with centrally located tumor recurrences treated with radical reirradiation and to provide preliminary guidance for appropriate patient selection.

## 2. Methods

### 2.1. Literature Search

Based on the Preferred Reporting Items for Systematic Reviews and Meta-Analysis (PRISMA) guidelines, a comprehensive literature search in Medline was performed using the following search terms: (re-irradiation) OR (reirradiation) OR (re-radiotherapy) AND (lung cancer). The database was accessed on 30 March 2021. Investigations published before 2000 or available in abstract form only as well as in languages other than English, French, and German were excluded. Two investigators (BG and FZ) performed the two-step selection process described in Figure 1 and the data extraction. First, the papers were selected by title and abstract. Secondly, the references in full papers were screened and—if suitable—included in this review. Studies were included as full papers in this review based on five criteria: (1) At least 66% of the patient cohort had NSCLC. This cutoff was chosen based on an epidemiological criterion. The incidence of NSCLC comprising adenocarcinoma and squamous cell carcinoma, is 60–70% [20,21]. According to Siegel et al. the 2-year survival rate for patients diagnosed with loco-regional NSCLC during the past decade ranges between 50% and 81% [22], which would result in a nominal mean of 65.5%. Based on the assumption that patients with a 2 years survival have NSCLCs with less malignant biology as their disease recurs intra-thoracically, we hypothesized that this proportion of 66% might benefit most from curative intent thoracic re-irradiation. (2) Similar to another review, thoracic re-irradiation with curative intent had to be administered with a minimal total physical dose of 50 Gy [8]. Alternatively, a median cumulative EQD_2_ > 100 Gy had to be achieved in both courses of treatment. (3) In both irradiation courses modern radiation techniques, i.e., 3D radiotherapy, IMRT/VMAT, protons, SABR, had to be used. (4) More than 50% of the patients were treated for centrally located lesions [10]. Studies on patient cohorts with mainly chest wall relapses, vertebral metastases, superior vena cava syndrome, and radiotherapy after surgery were excluded. (5) Similar to the review by Hunter, the study cohort had to have a minimum size of 30 patients [12].

### 2.2. EQD_2_

Since the radiation regimens were very heterogeneous, the physical doses were converted to EQD_2,_ with an α/β = 10 assumed for both tumor and acute effects, *D* as total physical dose and *d* as dose per fraction.
$$\mathrm{EQD}2=D\times \;\frac{d+\alpha /\beta }{2+\alpha /\beta }$$

### 2.3. Toxicity

As mentioned before, a major problem of thoracic re-irradiation is toxicity and its inter-study comparability. Most of the included studies referred to the 4th version of the CTCAE classification except for Oghuri [23] and Grambozov [24], who used versions 3 and 5, respectively. With respect to the main OARs, i.e., lung and esophagus, these various editions of the CTCAE are similar. Low grade toxicity may be underreported as the studies included in the current analysis, were—with one exception [3]—retrospective. In the majority of studies, toxicity was categorized according to the time of occurrence (acute versus late), with a distinction being made between low (G1–G2), high (G3–G4), and lethal.

### 2.4. Statistics

The numbers for mOS, mPFS and mLRC were extracted from the Kaplan–Meier plot, if they were not stated in the publications. After weighting the studies according to the number of patients included, clinical endpoints and toxicity were correlated to treatment-related parameters such as PTV, total dose, cumulative EQD_2_, and systemic treatment using the Pearson correlation.

## 3. Results

### 3.1. Selected Studies

The literature search using the above-mentioned search terms retrieved 227 studies. After excluding 191 studies for the reasons described in Figure 1, 36 papers were assessed for eligibility. After full-text review, 25 studies were excluded for one or more of the following reasons: (a) the number of included patients was less than 30, (b) more than 50% of the patients had peripheral tumors, and (c) more than 50% of the patients received palliative treatment. Finally, 11 studies were included in the analysis (Figure 1). The selected studies were published between 2012 and 2021 (Table 1 and Table 2), 1 of which was prospectively designed [3]. The median number of patients per study was 46 (range: 30–102) adding up to 524 who were included in this analysis. The median proportion of centrally located tumors was 74% (range: 52–100%). Most of the patients 514/524 (98%) finished the re-irradiaton course as scheduled. As for histology, the minimum proportion of NSCLC patients was 72% [25]. Although in 9 studies at least 66% of the patients had a performance score of 0–1, this was not specified in one study [26] and another summarized all patients with ECOG 0–2 [3]. All included studies listed the disease stages in detail-either at RT or at Re-RT. Locally advanced lung cancer, i.e., UICC IIb to IIIc, was present in more than 50% of the patients. Schlampp’s study represented a borderline case that was included because the cohort consisted of 62 patients with initial stage III tumors who received an average EQD_2_ of 99.5 Gy in both RT courses [27].

### 3.2. Clinical Outcome: mOS, mPFS, mLRC

The mOS was 18.1 months (range 9.3–25.1), with five studies above this median (Figure 2) and large ranges reaching up to 76.9 months [28]. The mPFS was 9 months (range: 4.5 [29]–16 [25]) and the mLRC was 12.1 months (range: 6.5 [27]–20 [30]). The median proportion of long-term survivors after re-irradiation was 39% (range: 15% [27]–50% [26]). Outcome data are summarised in Table 3.

### 3.3. Toxicity

As for toxicity, the current review focuses on side effects originating from the oesophagus, lungs, heart, vessels, and spinal cord. The large variety of reported side effects was divided into acute, late, and grade 5 toxicity (Table 4). Severe (grade 3–4), acute, and late toxicities were observed in 0% [30]–39% [3] and 0% [25,29] to 12 % [3], respectively. Seven studies reported therapy-related mortality between 2% [24,27] and 14% [26].

### 3.4. Prognostic Factors

Prognostic and predictive variables for both clinical outcome (OS, PFS, LRC) and main toxicities, i.e., esophageal and pulmonary, are summarized in Table 5. If only the five studies that included multivariate analysis are considered [23,24,26,28,31], in-field relapse combined with central tumor location [28] seemed to influence toxicity (*p*-value 0.03 HR 6.4). The following predictors and prognosticators for OS could be identified: PTV (*p*-value 0.000 HR 1.007) [24], histology (*p*-value 0.004 HR 0.4; *p*-value 0.04 HR 3.5) [11,23], cCRT (*p*-value 0.0045 HR 2.6 [11], re-irradiation dose (*p*-value 0.021 HR 0.2) [11], recurrent tumor size (*p*-value 0.001 HR 17.3) [23] and ECOG (*p*-value 0.028 HR 2.5) [11]. T4-stage (*p*-value 0.013 HR 3.5), cCRT (*p*-value 0.027 HR 0.5) [11] and extrathoracic disease (*p*-value 0.02 HR 2.9) [26] were found to predict PFS. Furthermore, LRC was influenced by cCRT (*p*-value 0.004 HR 6.5) [11], interval (*p*-value 0.012 HR 0.4) [11], histology (*p*-value 0.03, HR 3.7) [23], and recurrent tumor size (*p*-value 0.04 HR 0.2) [28].

### 3.5. Treatment

#### 3.5.1. Re-Irradiation

The majority of patients were treated with advanced technologies for re-irradiation such as IMRT, SABR, and protons (Table 2). Only Ohguri and co-workers used 3D conventional radiation technique [23]. The median re-irradiation PTV and dose were 96 mL (range: 13–248 mL) and 55 Gy (40–67 Gy), respectively. The median cumulative EQD_2_ was 124 Gy (range: 100–209 Gy) excluding the prospective study by Chao, in whose cohort summary treatment plans could not be calculated due to a lack of data on the first treatment course, as the patients were treated in external centers [3]. The median total dose at re-RT was 56 Gy (range: 50 to 64 Gy). Since this parameter was reported in all studies, the correlations between dose and clinical endpoints were calculated with this variable. The total dose at re-RT correlated with both LRC (Figure 3a, *p*-value = 0.012) and OS (Figure 3b, *p*-value = 0.007) with a close relation between these two clinical endpoints (Figure 3c, *p*-value = 0.006). The occurrence of acute toxicity grade 1 to 4 was dependent on the PTV size at re-irradiation (Figure 4, *p*-value = 0.033). In two studies [23,25] the re-irradiation PTV was not given three-dimensionally; therefore, these two studies were excluded from this correlation calculation. Reports on doses to the OARs were inconsistent. One study [24] explicitly mentioned constraints, whereas another [27] listed the actual doses administered to the OARs. With regard to the four main OARs (esophagus, lung, heart, spinal cord), which limit doses in thoracic irradiation, the following information could be extracted from the eleven papers included in the analysis. (1) For the esophagus cumulative D_max_, V60 and MED ([11,23,27] were considered with a limit set at a cumulative D_max_ of 100 Gy [24] and a reported D_max_ of 92.5 Gy [27]. (2) For the lungs, the following parameters were used for limiting pulmonary toxicity: V5, V10, V20, MLD [11,23,24,27]. In Grambozov’s study, the V20_total lung_ was set at 50% [24]. Furthermore, a V20_total lung_ of 31 Gy, MLD of 18.6 Gy and V5 of 80.3% was reported in a study by Schlampp [27]. Regarding cardiac toxicity, V40 and D_max_ were used as dose parameters in McAvoy’s study. Moreover, in one study, a MHD of 7.9 Gy was reported [27], and in another a constraint of V20 < 20% [24] was used. The cumulative dose limit for the spinal cord dose was D_max_ 75 Gy [24].

#### 3.5.2. Chemo-Immunotherapy at Re-Irradiation

Apart from the patients in the studies by Kilburn [25], Ohguri [23], and Chao [3], most patients received sequential CT at re-irradiation. Moreover, in two studies, 21/47 (45%) and 6/50 (12%) of the patients were additionally treated with immunotherapy, respectively [26,32]. The addition of one of the following ICIs to chemotherapy and re-irradiation in the treatment of in-field thoracic recurrence has been associated with prolonged OS [32]: atezolizumab, pembrolizumab, nivolumab, or durvalumab [32]. The potential synergistic effect of radiation treatment and immunotherapy in relation to local and systemic anti-tumor response, which could possibly be responsible for the above-mentioned result, is currently a topic of intensive discussion.

## 4. Discussion

Although the evidence for thoracic re-irradiation is mainly based on retrospective studies, this review revealed pertinent results with respect to curative intent re-RT for centrally located tumor recurrence. At the present stage, the achievable mOS after re-RT is approximately 1.5 years with the potential for long-term survival (Figure 2 and Table 3). Treatment-related parameters such as radiation dose and volume correlate with clinical outcome and toxicity (Figure 3a–c and Figure 4).

As mentioned above, the probability for isolated local failure is 25% [3,4]. Since chemotherapy shows response rates of only 10–20% [27] and these patients are often regarded as inoperable, re-RT remains the mainstay of therapy. In this context, a major issue is the geometrical definition of the locally recurrent tumor. This is even more challenging with centrally located tumors, which cannot easily be distinguished from mediastinal soft tissue structures. The importance of exact delineation of the recurrent tumor GTV is highlighted by loco-regional relapse rates of 40% after re-RT [11,29]. Of note, in almost 40% of the patients, the gross tumor would have been located outside the PTV if only CT had been used for treatment planning [33]. Therefore, based on the 7-step process described by Hunter [12], the use of PET-CT for restaging patients with central tumor recurrence appears to be indispensable. This requirement was met by the majority of the studies included in our analysis [11,14,15,24,25,28,29,30]. As shown by the current analysis, mLRC correlates with mOS (Figure 3c), which is in line with the studies by McAvoy showing that patients with in-field relapse had poorer survival [11,29]. This is also corroborated by a Dutch study with 13.5 months mOS on average, which drops to 6.3 months in the case of local progression [14,15]. Long-term survival, i.e., >24 months, can be achieved in approximately 40% of cases with centrally located recurrences (Table 3). Similar results have been reported by Okamoto, who published data from eight long-term survivors [34]. The comparatively low mOS rate of 9.3 months reported by Schlampp, on the other hand, could be related to the moderate total re-irradiation dose of 39.5 Gy [27]. In fact, only two patients received up to 60 Gy total dose during the second course of radiation [27].

In these elderly and often fragile patients, toxicity is paramount. Side effects were generally rated as mild, with two studies explicitly indicating that they were not higher than would have been expected with primary treatment [11,27]. Treatment-related mortality, however, reached up to 14% [26], which was caused by radiation-induced damage to the lung and blood vessels [14,15,25,26,27,28]. Of note, in some cases it may have been difficult to differentiate between treatment-related toxicity and tumor persistence or progression that may have led to vessel erosion [14,15,30]. High grade (G3–4) esophageal toxicity with up to 9% [29] resulted from high composite radiation doses to the mediastinum. Likewise, the proton-study by Chao reported 39% severe side effects including esophagitis [3]. Hence, it seems that even with high end technology such as protons [3,11,29] and IMRT [14,15,24,26,27,30], high grade toxicity that affects quality of life is—to some extent—unavoidable. In this respect, the systematic review by Maddalo highlighted the heterogeneity of the cohorts including patients treated palliatively and scored with different toxicity systems such as CTCAE, RTOG/EORTC, and WHO [35]. Although this hampers the inter-study comparability, the discretion of the treating physician may also contribute to inconsistencies that preclude establishing reliable NTCP models. Of note, the current analysis is more consistent with respect to toxicity scoring than the meta-analysis by Maddalo since 9/11 (82%) studies use CTCAE version 4, whereas the other two report side effects by means of versions 3 [23] and 5 [24].

In the context of patient selection, a number of parameters for outcome prediction are currently discussed. The issue is further complicated by the diversity of statistical approaches and the lack of reliable predictive models generated from prospective trials. As shown by our review only a minority of the studies included a multivariate analysis (Table 5). For mOS, mPFS, and mLRC, major therapy-related parameters found in MVA were tumor size and PTV at re-RT [23,24,28], radiation dose [11], cCRT [11], and time interval between treatment courses [11]. Tumor-related parameters relevant for outcome were histology [11,23], and extrathoracic disease [26]. Toxicity depended on in-field recurrence combined with central location [28] and PTV size (Figure 4).

The current analysis should be seen in connection with two comprehensive reviews from 2014 [5] and 2020 [36] although these reports are only partially comparable to ours, since they included patients with very low volume disease who could be treated with SABR. In keeping with published literature [5], curative intent re-irradiation must be compared with palliative therapy, either by RT, which reaches an OS of approximately five months [5], or systemic treatment with an OS of 10–12 months [5]. The results of the re-irradiation studies summarized in the current overview compare favourably with both approaches. Although the clinical situation on which this review focuses, i.e., centrally localized tumor recurrences, is more challenging than peripheral recurrences, the 18.1 months mOS of the current analysis are slightly higher than the 17 months of the above-mentioned review by de Ruysscher [5]. The very same paper included patients who could be treated with SABR, meaning that they had peripheral tumor recurrences without hilar or mediastinal lymph node disease. The rate of side effects was also in the same range as in the mentioned review, although the treatment of centrally located recurrences is likely to result in higher toxicity due to their proximity to hilar and mediastinal structures. In addition, grade 3 to 4 pulmonary and esophageal toxicity were 0–21% and 0–9%, respectively [5], compared with 0–26% and 0–9% in the present review (Table 4). As the review by Nicosia included four studies with early-stage cancers, reported toxicity of 6.8% and 0.7% for grade 3–4 and grade 5, respectively, was lower than in the current analysis, whereas the summary ranges for mOS and mLRC were similar [36].

Since reliable predictive models are missing and DVH data are mostly unavailable, dose constraints for thoracic re-irradiation remain undefined. In this respect, the study by Grambozov [24], which incorporates previous experiences [34,37,38,39,40] and the review by de Ruysscher [5] may provide some preliminary guidance for normal tissue tolerance. The cumulative D_max_ for the esophagus, trachea, and spinal cord were set at 100 Gy, 110 Gy, and 75 Gy, respectively. The limits for lung and heart were as follows: V20_total lung_ < 50% and V25_heart_ < 20%. With these constraints, high-grade (G3–4) late toxicity and treatment-related mortality amounted to 2% each [24]. In order to avoid lethal bleedings, the D_max_ to the major central vessels should not exceed 120 Gy in more than 1 mL volume [5,31]. At present, dose cutoffs are extant for patients who received SABR as one [41] or multiple treatment courses [42,43,44], which is only of limited value in the current context because centrally located tumors are usually not amenable to SABR. In a recent single center evaluation of 21 NSCLC patients, the authors described toxicity as acceptable with MLD of 10 Gy and oesophagus D0.1 of 62 Gy [45]. The very same study also included 21 individuals with one of the following diseases: SCLC, esophageal carcinoma, pleural mesothelioma, and metastases [45]. This again reflects the heterogeneous clinical reality, which constitutes a major problem in defining dose limits for OARs in thoracic re-irradiation apart from well-defined treatment situations with very limited intrathoracic tumor burden, in which SABR can be used.

To date, the available evidence on OS, toxicity and prognosis is derived from retrospectively analyzed patient cohorts, which precludes conclusive answers to the clinically decisive question of how to select patients for this potentially toxic treatment. With the exception of the only prospective study [3], none of the trials described a systematic patient selection process. Nevertheless, the following repeatedly discussed parameters can be listed, which characterize patients who presumably benefit most from thoracic re-irradiation: (1) good general condition expressed in low PS (ECOG < 2) and young age (<65 years); (2) small volume and long interval between treatment courses as a sign of less malignant disease [8]. Under these pre-requisites and with new radiation technologies such as IMRT/VMAT and protons in combination with state-of-the-art IGRT, the notion that high cumulative doses > 100 Gy may be too toxic for patients with centrally located tumors [8] seems be outdated. In order to shed light on optimal patient selection and treatment regimens, well designed prospective re-irradiation trials are necessary. Thus far, a clintrials.gov search yielded two clinical studies whose publications are still pending and an in silico trial [46]. With mOS rates of five months after primary CRT followed by Durvalumab in PDL-1 positive LA-NSCLC patients [6,47,48], it can be expected that the proportion of patients experiencing loco-regional relapse will rise. However, thus far the number of patients who received ICIs and/or TKIs in the re-irradiation setting was too small to be systematically analyzed. Immunotherapy is an emerging scientific field of high clinical relevance. One of the main confounders in regard to the efficacy of immunotherapy seems to be gender. Pooled data from two large meta-analyses with 11.000 and 12.700 patients, respectively, report conflicting results: although Conforti reports a gender difference [18], Wallis states that males and females benefit equally from immunotherapy [49]. Other factors that may bias the efficacy of immunotherapy are alcohol consumption, exercise, obesity, circadian rhythm, psycho-emotional stress, smoking, and race (reviewed by Deshpande [50]). As detailed information on these parameters is missing in the studies presented in this review, the impact of these confounders on patient outcome after thoracic re-irradiation combined with immunotherapy is a topic for future analyses.

This review is limited by the retrospective nature of most included studies. Although we tried to minimize the inter-cohort heterogeneity by focusing on curative intent thoracic RT for mainly centrally located tumors there is still little consistency mainly due to different re-irradiation dose concepts and techniques, which reflects the variety of patients in daily clinical routine. Nevertheless, this is the first review to focus on centrally located recurrences of NSCLC, which are often deemed unsuitable for re-irradiation.

## 5. Conclusions

Since the evidence on re-irradiation with curative intent for central recurrences of NSCLC is scarce and heterogeneous, definite conclusions cannot be drawn. Overall survival in this clinically challenging situation is expected to be about 1.5 years with dose and volume at re-irradiation as the most prominent factors influencing outcome and toxicity. As thoracic re-irradiation is a high-risk treatment, patients should be carefully selected based on good PS, younger age, small tumors, and longer intervals to recurrence. In this respect, prospective randomized studies are warranted.

## Figures and Tables

**Figure 1 cancers-14-00573-f001:**
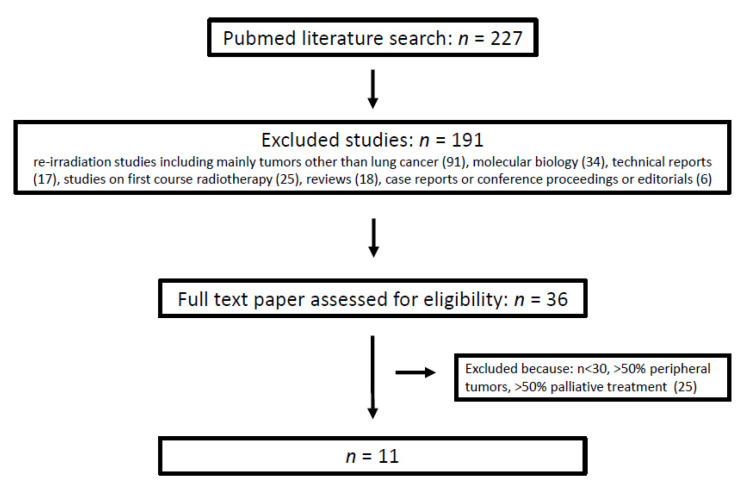
Study selection process.(This review was not registered with PROSPERO).

**Figure 2 cancers-14-00573-f002:**
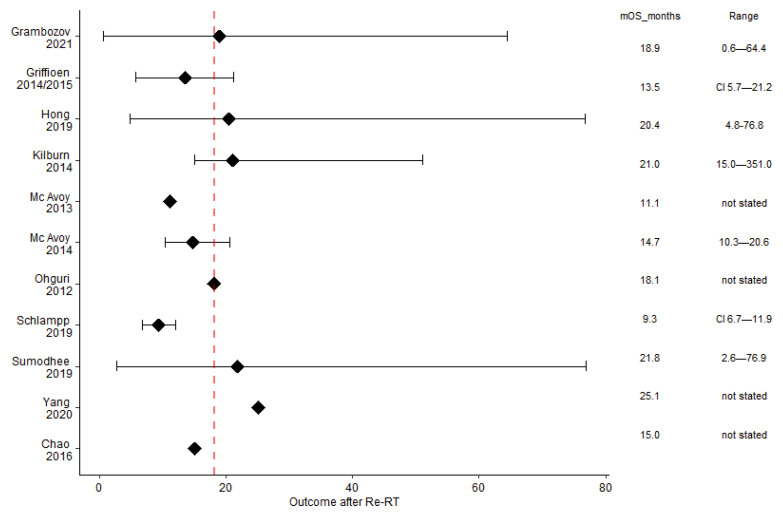
The median overall survival was 18.1 months (dashed red line). Although most studies presented the range, a 95% confidence interval was given only in two analyses. The black squares indicate the median OS (mOS).

**Figure 3 cancers-14-00573-f003:**
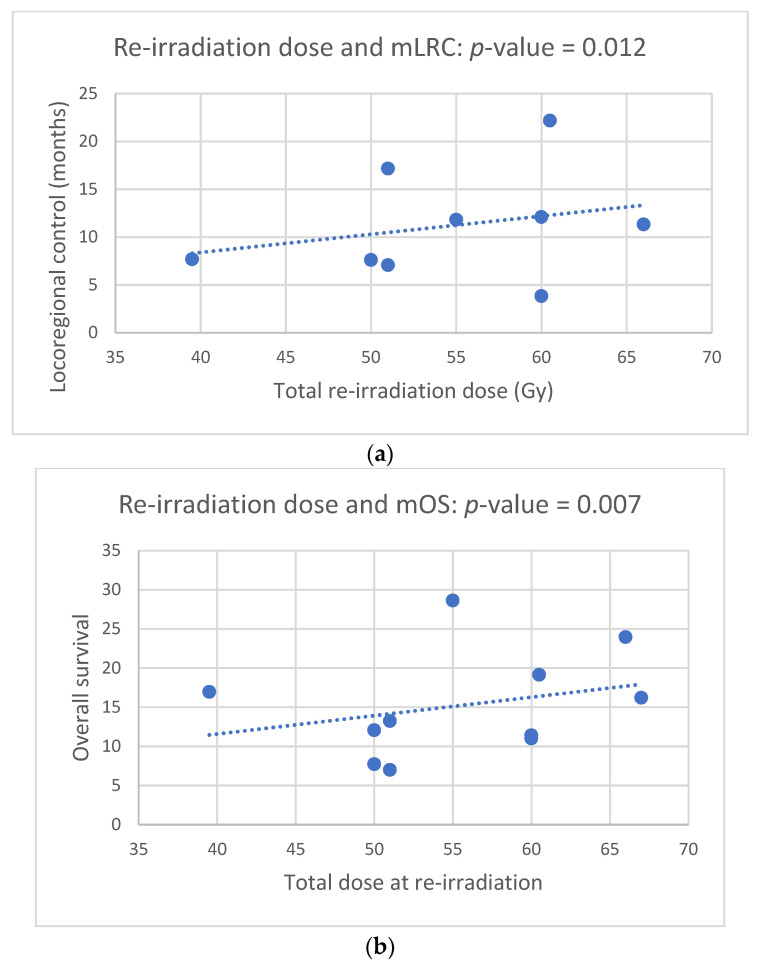
(**a**). Total dose at re-irradiation correlates to median loco-regional control (mLRC) (Pearson correlation, *p*-value = 0.012). (**b**). Total dose at re-irradiation correlates to median overall survival (Pearson correlation, *p*-value = 0.007). (**c**). Median OS correlates with median loco-regional control (Pearson correlation, *p*-value = 0.006).

**Figure 4 cancers-14-00573-f004:**
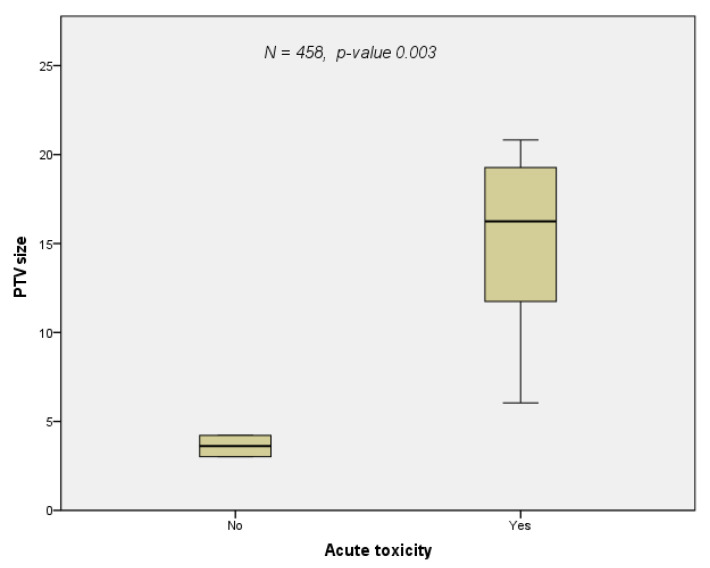
The occurrence of acute toxicity grade 1 to 4 versus PTV size at re-irradiation: patients who experienced acute toxicity grade 1 to 4 had a larger PTV compared with those without (Pearson correlation, *p*-value = 0.033). The studies by Kilburn and Ohguri could not be included in this calculation since they did not contain information on PTV size, hence the number of patients was reduced to 458.

**Table 1 cancers-14-00573-t001:** Patient and tumor characteristics in the selected studies (ns = not stated).

Study			Patient and Tumor Characteristics								
			Patients Included (N)	Median Age at Re-RT (years)	Re-RT Completed (N)	Histology	ECOG	Interval (Months)	Central Tumors (%)	UICC Stage at RT	UICC Stage at Re-RT
Retrospective	Grambozov	2021	47	66	47	NSCLC: 74% SCLC: 21% NOS: 5%	0–1: 85% 2: 15%	20	53	I: 15% II: 17% III: 53% IV: 15%	ns
	Griffioen (update Tetar)	2014/2015	30	63	30	NSCLC: 88% SCLC: 8% None: 4%	0–1: 83% 2: 17%	51	83	I–II: 12% III: 63% IV: 8% SCLC: 17%	I–II: 21% III: 63% IV: 8% SCLC: 8%
	Hong	2019	31	64	31	NSCLC: 74% SCLC: 22% NOS: 3%	0–1: 94%, 2: 6%	15	74	I: 10% II: 13% III: 45% IV: 6% NOS: 3% SCLC 22%	ns
	Kilburn	2014	33	64	33	NSCLC: 72% SCLC: 12% Mixed: 3% Other: 12%	0–1: 79% 2: 18% (1)	18	52	I: 31% II: 14% III: 45% IV: 10%	ns
	McAvoy	2013	33	69	31	NSCLC: 100%	0–1: 67% >1: 33%	36	85	I: 21% II: 15% III: 61% IV: 3%	ns
	McAvoy	2014	102	68	99	NSCLC: 100%	0–1: 81% >1: 19%	17	87	I: 29% II: 16% III: 45% IV: 9%	ns
	Ohguri	2012	33	68	33	NSCLC: 100%	0–1: 73% >1: 27%	8	58	I: 6% II: 12% III: 52% IV: 12% (2)	ns
	Schlampp	2019	62	63	62	NSCLC: 78% SCLC 16% NOS 6%	0–1: 100%	14	100	III: 100%	ns
	Sumodhee	2019	46	66	46	NSCLC: 100%	0–1: 100%	23	52	III: 100%	ns
	Yang	2020	50	65	50	NSCLC: 78% SCLC 8% NOS 14%	ns	13	86	ns	IIb–IIIc: 68% IV: 32%
Prospective	Chao	2016	57	65	52 (3)	NSCLC: 100%	0–2: 100%	19	61	ns	I: 21% II: 7% III: 62% IV: 11%

(1) One patient is missing in the patient table; it remains unclear what the ECOG in this patient was. (2) A total of 18% of the patient cohort received surgery. (3) Five patients did not complete re-RT because of disease progression or toxicity.

**Table 2 cancers-14-00573-t002:** Parameters regarding radiation technique, treatment volume and dose as well as systemic treatment.

Study			Treatment						
			Re-RT-Technique	Median PTV at RT (mL)	Median PTV at Re-RT (mL)	Median Total Dose at RT (Gy)	Median Total Dose at Re-RT (Gy)	Cumulative EQD2 (Gy)	Systemic Treatment at Re-RT
Retrospective	Grambozov	2021	IMRT	ns	47	74	51	131	sCRT (1): 57% none: 43%
	Griffioen (update Tetar)	2014/2015	IMRT	539	248	60	60	120	cCRT: 8% sCRT: 54%
	Hong	2019	IMRT: 68% SABR: 32%	353	51	64 (2)	55 (2)	119	cCRT:10% sCRT: 61% none: 29%
	Kilburn	2014	SABR	ns	2.5 cm	60	50	209 (3)	cCRT: 61% sCRT: 9% none: 30% (4)
	McAvoy	2013	Protons	ns	96 (5)	62	66	128	cCRT: 24% sCRT: 51% none: 25%
	McAvoy	2014	Protons SABR	ns	94	70	61	131	cCRT: 33% sCRT: 48% none: 19%
	Ohguri	2012	3D (6)	112 cm^2^	38 cm^2^	70	50	115	cCRT: 46% sCRT: ns
	Schlampp	2019	IMRT	459	176	60	40	100	cCRT: 0% sCRT: 92%
	Sumodhee	2019	SABR	ns	13	66	60	196	cCRT: 0% sCRT: 22% none: 78%
	Yang	2020	3D: 14% IMRT: 86%	529	202	60	51	106	cCRT: 18% sCRT (7): 42%
Prospective	Chao	2016	Protons	ns	108 (8)	ns	67	ns	cCRT: 68% sCRT: ns

(1) Immunotherapy was administered after RT in 21/47 (45%) of the patients. (2) These were the median total doses for NSCLC. (3) Composite plans were available from 19 patients. (4) Information on 23 patients was available. (5) ITV. (6) Hyperthermia in 46% of the patients during the first radiation course. (7) 6/50 (12%) of the patients received immunotherapy or TKIs. (8) CTV.

**Table 3 cancers-14-00573-t003:** Clinical outcome after re-irradiation (ns = not stated, nr = not reached). The outcome numbers in brackets and square brackets refer to crude ranges and 95% confidence intervals, respectively.

Study			Outcome after Re-RT			
			mOS (Months)	mPFS (Months)	mLRC (Months)	Patients Alive at 24 Months after Re-RT (%)
Retrospective	Grambozov	2021	18.9 (16.5–21.3)	ns	7.9 (6.7–9)	30
	Griffioen (update Tetar)	2014/2015	13.5 (5.7–21.2)	8.4 (5.5–11.3)	6.7 (2.5–11.0)	23
	Hong	2019	20.4 (4.8–76.8)	15.4 (3.4–76.8)	20 (1)	39
	Kilburn	2014	21 (2) (15–51)	16 (2) (6.6–nr)	not reached	45
	McAvoy	2013	11.1	4.5	18	33
	McAvoy	2014	14.7 (10.3–20.6)	11.4 (6.8–23.8)	11.4 (8.6–22.7)	33
	Ohguri	2012	18.1	6.7	12.1	45 (1)
	Schlampp	2019	9.3 (6.7–11.9)	ns	6.5 (6.0–7.0)	15 (1)
	Sumodhee	2019	21.8 (2.6–76.9)	9.6 (1–62.5)	13.8 (1–76.9)	45
	Yang	2020	25.1	5.9	18	50
Prospective	Chao	2016	15	14	ns	43

nr = not reached. (1) This number was read off the Kaplan–Meier-plot. (2) These numbers refer to the 25 non-metastatic patients only.

**Table 4 cancers-14-00573-t004:** Toxicity (ns = not stated).

Study			Toxicity		
			Acute	Late	Lethal
Retrospective	Grambozov	2021	G1-2: 9% (esophageal), 2% (pulmonary) G3-4: 4% (esophageal)	G1-2: 0%, G3-4: 2% (hemorrhage)	2% (cardiac)
	Griffioen (update Tetar)	2014/2015	G1-2: 88%, G3-4: 10%	G2: 21% (vertebral collapse, pulmonary)	13% (hemorrhage)
	Hong	2019	G1-2: 90% (pulmonary), 19%(esophageal) G3-4: 0%	G1-2: 49% (pulmonary) 6% (esophageal) G3-4: 3% (pericarditis)	0%
	Kilburn	2014	G1-2: 36% (pulmonary, chestwall pain) G3-4: 3% (pulmonary)	G1-4: 0%	3% (aorto-esophageal fistula)
	McAvoy	2013	G1-2: 36% (pulmonary), 36% (chestwall), 15% (esophageal), 15% (cardiac) G3-4: 21% (pulmonary), 9% (esophageal) 3% (cardiac)	G1-4: 0%	0%
	McAvoy	2014	G1-2: 18% (esophageal), 27% (pneumonitis) G3-4: 7% (esophageal), 10% (pneumonitis)	G1-2: ns G3-4: 1% (pulmonary)	0%
	Ohguri	2012	G1-2: 9% (pneumonitis), 15% (dermatitis), 6% (hematological) G3-4: 9% (thermal burns), 3% (pleuritis)	G1-2: 0% G3-4: 3% (brachial plexus neuritis)	0%
	Schlampp	2019	G1-2: 24% (including 19% pneumonitis) G3-4: 8% (pneumonitis)	G1-2: 21% (pulmonary) G3-4: 5% (esophageal, pulmonary)	2% (pulmonary)
	Sumodhee	2019	ns	ns	4% (pulmonary) (1)
	Yang	2020	G1-2: 22% (pulmonary) G3-4: 26% (pulmonary)	G1-4: 0%	14% (pulmonary)
Prospective	Chao	2016	G1-2: ns G3-4: 39%	G1-2: ns G3-4: 12%	11% (2)

(1) Additionally, 7% lethal lung infections are mentioned in the discussion. (2) Six patients with one of the following lethal side effects: hemorrhage, sepsis, anorexia, pneumonitis, respiratory failure, tracheoesophageal fistula.

**Table 5 cancers-14-00573-t005:** Prognostic and predictive markers for toxicity and clinical outcome.

Study			Esophageal Toxicity	Pulmonary Toxicity	OS	PFS	LRC	Statistics
Retrospective	Grambozov	2021	ns	ns	PTV (*p*-value 0.000, HR 1.007)	none	none	multivariate
	Griffioen (update Tetar)	2014/2015	ns	ns	PTV	PTV	ns	univariate
	Hong	2018	ns	MLD	EQD2 (Re-RT), cumulative EQD2 (both courses)	gender, CT after Re-RT, GTV, PTV, fraction size	EQD2 (Re-RT), cumulative EQD2 (both courses)	univariate
	Kilburn	2014	ns	ns	ns	ns	ns	ns
	McAvoy	2013	none	none	ns	ns	ns	univariate
	McAvoy	2014	none	none	histology (*p*-value 0.004, HR 0.4), cCRT (*p*-value 0.0045, HR 2.6), EQD2 at Re-RT (*p*-value 0.021, HR 0.2), ECOG (*p*-value 0.028; HR 2.5)	T4 (*p*-value 0.013, HR 3.5), cCRT (*p*-value 0.027, HR 0.5)	cCRT (*p*-value 0.004, HR 6.5), interval (*p*-value 0.012, HR 0.4)	multivariate
	Ohguri	2012	ns	ns	histology (*p*-value 0.04, HR 3.5), recurrent tumor size (*p*-value 0.001, HR 17.3)	none	histology (*p*-value 0.03, HR 3.7)	multivariate
	Schlampp	2019	ns	ns	nodal involvement, total Re-RT dose, dose to aorta, interval, FEV1	ns	none	univariate
	Sumodhee	2019	in-field relapse combined with central tumor (*p*-value 0.03, HR 6.4)	in-field relapse combined with central tumor (*p*-value 0.03, HR 6.4)	ns	ns	recurrent tumor size (*p*-value 0.04, HR 0.2)	multivariate
	Yang	2020	ns	ns	none	extrathoracic disease (*p*-value 0.02, HR 2.9)	none	multivariate
Prospective	Chao	2016	cCRT, MED, MHD, central volume overlap	cCRT, MED, MHD, central volume overlap	MED	none	none	univariate

(cCRT = concomittant chemoradiotherapy, CT = chemotherapy, EQD_2_ = biologically equivalent dose in 2 Gy fractions, FEV1 = forced expiratory volume in the first second, GTV = gross tumor volume, LRC = loco-regional control, MED = mean esophageal dose, MHD = mean heart dose, MLD = mean lung dose, ns = not stated, OS = overall survival, PFS = progression free survival, PTV = planning target volume, Re-RT = re-irradiation). In studies including multivariate analysis significance levels (*p*-values) and hazard ratios (HR) are given in brackets.

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
