# Peer review of "High Dose Thoracic Re-Irradiation and Chemo-Immunotherapy for Centrally Recurrent NSCLC"

_cancers, 2022, doi:10.3390/cancers14030573_

Round 1
Reviewer 1 Report
There is no problems other than most of research in this review is retrospective.
Could you write more clearly about it in abstract and main text?
Author Response
Reviewer 1
There is no problem other than most of the research in this review is retrospective. Could you write more clearly about it in abstract an main text.
Again, we would like thank reviewer 1 for her/his valuable comments on this paper. It is true that most of the evidence extant is retrospective (with the sole exception of the prospective proton study by Chao et al.). We apologize for being unclear in this respect. The manuscript was amended according to the reviewer’s suggestions by inserting the following sentences (green font, track changes) in the indicated sections of the text.
Abstract, conclusion
The evidence regarding curative re-irradiation for centrally recurrent NSCLC is primarily based on scarce retrospective data, which are characterized by a high degree of heterogeneity.
(…)
In this context, prospective trials are warranted to achieve essential advances in the field.
Discussion, last but one paragraph, first sentence
To date, the available evidence on OS, toxicity and prognosis is derived from retrospectively analyzed patient cohorts, which precludes conclusive answers to the clinically decisive question …
Conclusion
In this respect, prospective randomized studies are warranted.

Reviewer 2 Report
I have no further comments.
Author Response
Many thanks to reviewer 2 for considering our changes.
Best regards
Franz Zehentmayr

Reviewer 3 Report
I Congratulate the authors for providing the modifications. All my concerns are now addressed.
Author Response
We would like to thank reviewer 3 for accepting our answers!
Best regards
Franz Zehentmayr

This manuscript is a resubmission of an earlier submission. The following is a list of the peer review reports and author responses from that submission.
Round 1
Reviewer 1 Report
Of course, thoracic reirradiation is very important problem. However, there are few prospective studies, as authors say. So, if you write review for it, I agree to accept. I think this rewrite as review, not but systematic review.
Reviewer 2 Report
The authors have presented a paper about "Thoracic re-irradiation and chemo-immunotherapy for centrally recurrent NSCLC: a systematic review".
The topic is interesting and the authors have made a great deal of effort but I have some major concerns.
Why did the authors choose the following criteria "At least 66% of the patient cohort had NSCLC"? They should explain with a valid scientific criteria such choice.
Also in the discussion section I feel the manuscript could benefit from a literature update including the latest evidence.
Reviewer 3 Report
In the present systematic review, authors perform a systematic review on published articles on irradiation in immunotherapy treated lung cancer patients and infer the efficacy of curative radiation. I have several reservations, my comments are appended as below:
- Abstract-introduction- provide rationale statement.
- Reference 1: do the study provides statistical inference? In general, while citing research involving human patients, authors should annotate the statistical figures (HR, P-value).
- As immune checkpoints are the central theme, authors should first provide their basis and components in the introduction section. Authors may refer: PMID: 33076303, PMID: 34572799.
- Reference 8- elaborate on mutations.
- Were the patients included in the study uniformly treated with the same dose of radiation or there is heterogeneity?
- Toxicity- authors should elaborate on parameters to access toxicity.
- Do authors observe any gender bias in response to therapy?
- What kind of immunotherapy do the do studies include?
- Tables- annotate with statistical inference.
- Figure 4- indicate n and statistical inference.
- There are other cofounders who may affect immunotherapy efficacy. For instance, author s may refer to PMID: 33076303 and discuss in light of the present study.